# The Intersection of Human Disturbance and Diel Activity, with Potential Consequences on Trophic Interactions

Michael A. Patten[1,2]*, Jutta C. Burger[3¤], Milan Mitrovich[4]

**1** Oklahoma Biological Survey, University of Oklahoma, Norman, Oklahoma, United States of America, **2** Department of Biology, University of Oklahoma, Norman, Oklahoma, United States of America, **3** Irvine Ranch Conservancy, Irvine, California, United States of America, **4** Natural Communities Coalition, Irvine, California, United States of America

¤ Current address: California Native Plant Society, Berkeley, California, United States of America
* mpatten@ou.edu

**Data Availability Statement:** Data cannot be shared publicly because they are co-owned by multiple land owners and the Irvine Ranch Conservancy (IRC), but they are available from IRC for researchers who meet approval for data access.

## Abstract

Direct effects of human disturbance on animal populations are well documented across habitats, biomes, and species, but indirect effects of diel have received less attention. An emerging field in applied ecology involves behavioral avoidance of or attraction to humans and their trappings. We posit trophic consequences, in terms of relative risk, for four species of mammals, each of which strongly avoids human activity, in urban reserves of coastal southern California. Two species, one predator and one prey, avoid human activity via a temporal shift to become "more nocturnal"—the species' activity is centered near dawn on days without human activity but nearer to midnight on days with human activity. Diel shifts have brought the species into greater overlap, respectively, with a key prey and a key predator, overlap that may increase encounter rate and thus increase relative risk of predation, with potential consequences for trophic dynamics and cascades: increased risk of predation may depress prey population, either directly (e.g., mortality) or indirectly (e.g., "landscape of fear"). Human use of reserves, especially in high population density regions, needs to be reconsidered either to reduce access or to restrict access entirely to areas that may provide refuge to both predators and prey.

## Introduction

Predation is a principal force that shapes an ecosystem. Predator–prey dynamics are complex; many pieces must fall into place before any individual is attacked and consumed by a predator. In this light predation can be thought of as a multi-stage process, the stages being encounter between prey and predator, active choice of prey species by a predator, and success of any given attack [1]. Because prey pay the ultimate price, it is expected that prey reduce their exposure to predators, and there is a good deal of evidence in support of this expectation [2]. Key anti-predator defenses include a decrease in activity or an increase in time spent in hiding [3]. Seclusion in a refuge need not be spatial but may be temporal.

Contact Yi-Chin Fang (yfang@irconservancy.org) for access requests.

**Funding:** This work was funded by Local Assistance Grant #P1482109 awarded by the California Department of Fish and Wildlife. https://www.wildlife.ca.gov/Conservation/Planning/NCCP/Grants The funders had no role in study design, data collection and analysis, decision to publish, or preparation of the manuscript.

**Competing interests:** The authors have declared that no competing interests exist.

Humans can disrupt natural predator-prey dynamics. Animals respond to human presence sometimes positively [4] but usually negatively [5–6]. Avoidance is a common response, particularly avoidance of areas with high levels of human activity or infrastructure [7–9], and avoidance can have the same ecological effects as predation [10–11]. Wildlife response also is affected by the type of activity; for example, domestic dogs that accompany hikers have an additive effect on wildlife displacement relative to hikers alone [12], and bicyclists plus hikers have a comparable additive or interactive effect relative to hikers alone [13].

Yet an animal cannot avoid an area of high human activity if accessible alternative habitat does not exist [6], a problem exacerbated in regions that have been developed extensively. In coastal southern California, for instance, human population continues to grow and so habitat loss continues apace. In this region wildlife response tends toward avoidance of areas of high human activity or heavy urbanization [14–17], with some species displaced temporally: they are more active at night is areas of high human activity [14–15,18]. Such temporal shifts have been documented in other regions [19–21] and could be ubiquitous in areas with high human activity [22].

Although diel or temporal shifts in activity in response to humans are well documented [22–25], much of this documentation comes from single-species studies or from multi-species studies in which each species was considered serially rather than in parallel. Moreover, in many cases shifts have been predicted or documented only for predators [26–27], even though various studies have shown that prey species (typically primary consumers) shift their activities in time or space to avoid humans [21,28–29]. It is reasonable to conclude that any species not a human commensal (or otherwise disturbance tolerant) will alter its behavior if humans are present. An outstanding question is not whether species shift their use of time or space in response to human presence but how such shifts alter trophic interactions and dynamics. Kuijper and colleagues [30] metaphorically referred to this effect as the "human shadow" cast on how predators and prey interact, and Magle and colleagues [29] speculated that human activities altered predator–prey dynamics when predator and prey were relegated to share habitat patches as a result of human encroachment.

An unanswered question is whether anthropogenic-induced diel shifts alter predator–prey dynamics, specifically in terms of altering encounter rate. Because "factors that increase a prey's encounter rate. . . should have a positive effect" on predation rate [3], high levels of human activity could, in principle, depress further recruitment levels depressed already by habitat loss and fragmentation. Our key research goal, then, was to lay groundwork to study the extent to which human disturbance alters the probability of encounter between prey species and their predators. Groundwork was provided by a large set (nearly 90,000 records) of camera-trap data collected across eight years at >50 stations situated in coastal southern California. We examined patterns across seven mammal species but focused on two pairs of predator and prey, the PUMA (*Puma concolor*) and the MULE DEER (*Odocoileus hemionus*), its principal food in the region [31], and the COYOTE (*Canis latrans*) and the GRAY FOX (*Urocyon cinereoargenteus*), of which the COYOTE is behaviorally dominant and a frequent predator [32–33]. We assessed not only diel shifts but further explored how encounter rates might change and speculated on the consequences of any a change.

## Methods

### Study area

The study was conducted in Orange County, California, within a complex of urban and urban-adjacent preserves once privately owned and now publicly owned by the County of Orange, City of Irvine, and City of Newport Beach [17]. Parks in our study span ~12,000 ha

(~30,000 acres) in central and coastal Orange County and are protected further through either the Orange County Central and Coastal Natural Community Conservation Plan (NCCP) or by conservation easements, as well as deed restrictions, park abandonment ordinances, and other legal mechanisms. These lands are adjacent both to a heavily urbanized landscape, which supports over three million people, and near to ~28,300 ha (70,000 acres) of U.S. National Forest. Public access is not only permitted but encouraged, chiefly via a managed access program with permit-only entry, regular docent-led programs, and monthly self-guided wilderness access days. Dogs are not permitted on managed-access lands. Climate is Mediterranean, and habitats consist of coastal sage scrub, chaparral, grassland, and oak woodland, with some ephemeral streams and permanent water sources (natural and man-made).

## Data set

Fixed location digital wildlife cameras (Cuddeback Expert 3300, Non Typical, Inc., Green Bay, Wisconsin, and HCO Scoutguard SG-565F, HCO Outdoor Products, Norcross Georgia) were installed across the landscape to monitor human and wildlife activity long term in the reserve system (S1 Table, S1 Fig). Cameras were installed and monitored as part of a reserve-wide wildlife and human activity monitoring program managed by the Irvine Ranch Conservancy long-term land management agreements with the following land owners: The Irvine Company (no contract number), Orange County Parks (MA-012-13012148), City of Irvine (MA-4950), and City of Newport Beach (contract 03012008). Additional access and implementation permits were not needed because monitoring activities were encompassed in the scope of work for each management agreement. All landowners and federal and state agencies associated with the project team knew of and supported the project as a tool to inform adaptive management of wildlife and human activity in the NCCP and the adjacent reserve system.

Cuddeback cameras were used from 2007–2011, Scoutguard from 2012–2015 (the former were no longer available commercially). Different makes and models of cameras respond to stimulus in different ways, which could affect estimates of population size or occupancy, but our analyses and comparisons were relative (e.g., averages within-camera captures), so a correction factor was unnecessary. Cuddeback cameras had a flash range of 18 m, were triggered instantly by motion (6–30 m distance) and heat, and a had narrow field of view of 2 m at 10 m distance; Scoutguard cameras had a flash range of 15 m, were triggered at 2–10 m, and had a wide detection angle of 52˚. Sensitivity of each camera was adjusted to maximize probability of species detections but minimize superfluous photographs of moving vegetation or shadows. Each camera was set for a 1-min delay between photographs to minimize duplicates of the same individual. Images were stamped with date and time on 1-GB compact flash cards, which were collected every two weeks. For each identifiable photograph, date, time, species detected, number of individuals, trap location, and notes were entered into the relational biodiversity database program Biota 2.04Ⓡ.

Data from 50 cameras were analyzed from June 2007–March 2015, although not all cameras operated continuously over that period. Thirty cameras were sited along trails, six adjacent to water troughs or water sources, and the remainder off-trail. Trail cameras were positioned obliquely (i.e., not perpendicularly) to improve capture rates of humans and wildlife, and all cameras were sited to maximize probability of detection and to provide a representative sample of managed urban-adjacent wildlands (for a map of camera placement, see ref. 17). Twenty-eight cameras recorded activity continuously since June 2007. If a human, bicycle, or vehicle was detected repeatedly within 1 h then all such photos in that hour were summed as a single record, chiefly to avoid over-counting people and their accouterments. If a wildlife species occurred repeatedly within a 5-min period, then photos were tallied as a single record;

only ~15% (4792 of 30519) of mammal detections, species for species, occurred >5 min. and <1 h, and reducing the data set to a 1-h cutoff did not materially change results (S2 Fig).

## Analyses

Diel patterns of seven species—the Bobcat (*Lynx rufus*), Puma, Gray Fox, Coyote, Striped Skunk (*Mephitis mephitis*), Northern Raccoon (*Procyon lotor*), and Mule Deer—in relation to human activity were analyzed using the full data set from June 2007–March 2015, which yielded 87,881 photos identifiable to species or activity. We considered all human activities— hiker, bicyclists, equestrian, vehicle, or domestic dog (despite prohibitions, many were detected, typically accompanied by a person on foot)—to constitute anthropogenic disturbance; in the study area, effects of each activity differed but all are negative [17]. We classified each focal mammal occurrence as either "undisturbed" or "disturbed," the latter defined as human activity at the same camera <24 h prior to a focal mammal detection. (We also defined disturbance in two other ways, as human activity <12 h prior or human activity between noon and sunrise the following morning, but results were consistent irrespective of definition. In no case did we consider human presence *after* a mammal detection; i.e., a human had to have been detected first.)

We analyzed daily activity using circular statistics, the "circle" being a 24-h clock. For each species we used Rayleigh's test to determine if activity departed from a null of random occurrence across a day. Circular analogs to ANOVA have restrictive assumptions that our data did not meet, so we used a non-parametric (and hence conservative) rank sum test of difference in mean angle to compare mean activity on undisturbed vs. disturbed days [34]. We used SAS 9.3 (SAS Institute, Cary, N.C.) for the rank sums test and package 'circular' in R or a spreadsheet for other tests. We quantified extent of activity overlap between species A and species B in undisturbed vs. disturbed conditions. To do so we used the polygon tool in ImageJ shareware to estimate shared activity (i.e., A ∩ B) against total activity (A U B). The quotient of the intersection and union is an estimate of proportion of overlap. We supplemented this simple descriptive step with a Bayesian estimate of a statistic, *d*, the trigonometric relationship at the heart of the Watson–Wheeler two-sample test of difference in mean vector, the logic of which is described and illustrated in Batschelet [34]. In short, the statistic assess whether the mean angle for species A differs from the mean angle for species B by comparing their sum to a composite angle (a "grand mean" vector). Our JAGS code to estimate *d* used a von Mises process for the likelihood and flat priors for mean angle.

We express encounter between predator and prey as odds ratios—which for these data is a measure of relative risk—of joint probability of disturbed vs. undisturbed occurrences binned at 15-min intervals. For any given species, the probability of occurrence in an interval is the quotient of the sum of occurrences in that interval and the total occurrences of that species. The joint probability of occurrence of predator and prey is the product of probabilities per interval, with overall joint probability the sum of all within-interval joint probabilities (i.e., the odds of two odds ratios). In this way we calculate a joint probability for undisturbed ($p_u$) and disturbed ($p_d$), with the odds ratio estimated as $[p_d \cdot (1- p_u)]/[p_u \cdot (1- p_d)]$. We estimated 95% Bayesian credible intervals (CI)—in JAGS and R, with a Bernoulli process for the likelihood— for the odds ratios to assess statistical significance [35]; i.e., we determined whether the odds ratio differed from 1.0 (i.e., no relative risk) by means of whether the CI overlapped 1.0.

The predator–prey system of the Puma and Mule Deer is coupled tightly, so we focused on it to examine potential effects of diel shifts. We plotted annual estimated abundance from an *N*-mixture model (package 'unmarked' in R) of each species and of human disturbance. An *N*-mixture model is, in effect, an occupancy model wherein the response variable is abundance

rather than presence/absence, with an occupancy model being an approach to estimate the probability of occupancy ($\Psi$) conditional on the probability of detection ($p$). We can extract from an $N$-mixture model both an estimate of abundance and an estimate of $\Psi$, both of them adjusted for how easy or difficult it is to detect a species. Annual data for the $N$-mixture model were detections per quarter (March, June, September, and December) per camera-day, such that each year had only two (2016), three (2007), or four (2008–2015) "repeat" surveys per camera. If a diel shift increased encounter odds, then we predicted deer detections to decrease while Puma detections would be stable or increase. We explored this prediction both via a plot of mean estimated abundance across years and via slopes from a Bayesian beta regression of $\Psi$ against year (in JAGS and R, with flat priors, a burn-in of 100,000, and MCMC iterations of 1,000,000 thinned by 1000). Lastly, we calculated expected versus observed joint occurrence, meaning detections of both a deer and a Puma at the same camera on the same day, to infer whether deer co-occur as often as expected if their presence was independent of the Puma's presence.

## Results

As expected, human activity was concentrated between mid-morning and noon, with ~95% of activity during daylight hours (S3 Fig). Disturbance varied among cameras: the proportion of days with human presence averaged 0.788 (SD $\approx$ 0.285, estimated from the weighted mean reported here), with a range of 0.128 to 1.000. With regard to the maximum proportion, human presence was detected daily at two of the fifty cameras, although neither of these cameras was operated particularly long (31 days and 165 days).

Four of the seven mammal species for which we had sufficient data shifted their temporal distribution in response to human presence (Table 1, S4 Fig). Shifts were particularly striking for two species, the Coyote, from shortly before dawn to shortly before midnight (Fig 1), and the Mule Deer, from around sunrise to shortly after sunset (Fig 2). Marked diel shifts in these two species increased overlap from ~⅓ to ~½ with their respective interactors in predator–prey dynamics, the Gray Fox and the Puma (Figs 1 and 2), neither of which shifted temporally (Table 1). Under undisturbed conditions mean activity of the Coyote and fox differed ($d$ = 148 with Bayesian 95% CI of [125, 171]), but under disturbed conditions mean activity did not differ ($d$ = 1 [–14, 16]). Likewise, under undisturbed conditions mean activity of the Puma and deer differed ($d$ = 222 [122, 317]), but under disturbed conditions mean activity did not differ ($d$ = 11 [–4, 26]). Each diel shift increased the relative risk of predation of both the Gray Fox (odds ratio [95% Bayesian credible intervals]: 1.39 [1.14, 1.64]) and the Mule Deer (1.31 [1.06, 1.56]).

**Table 1. Temporal distribution across undisturbed and disturbed days for seven mammal species in coastal Orange County, California.** All means (from angle θ, with it and 95% confidence intervals, CI, converted to time of day, $t$, on a 24-h clock) differed significantly (Rayleigh test: $P$ < 0.0001) from a random distribution of occurrences around the clock. Statistic $r$, which varies from 0 to 1, is a standardized measure of concentration at the mean. At the right, results are presented a rank sum test of temporal shift between undisturbed vs. disturbed occurrences.

| | undisturbed | | | disturbed | | | | |
|---|---|---|---|---|---|---|---|---|
| species | mean $t$ | $r$ | $n$ | mean $t$ | $r$ | $n$ | $Z$ | $P$ |
| Lynx rufus | 00:24 | 0.20 | 2140 | 22:44 | 0.42 | 992 | -3.53 | <0.01 |
| Puma concolor | 00:15 | 0.40 | 514 | 22:58 | 0.59 | 157 | -1.36 | 0.17 |
| Urocyon cinereoargenteus | 00:34 | 0.60 | 971 | 23:10 | 0.70 | 268 | -1.42 | 0.16 |
| Canis latrans | 04:04 | 0.26 | 3671 | 23:08 | 0.40 | 2440 | -9.04 | <0.01 |
| Mephitis mephitis | 01:30 | 0.61 | 280 | 00:52 | 0.74 | 108 | -2.06 | 0.04 |
| Procyon lotor | 00:23 | 0.55 | 599 | 23:59 | 0.62 | 91 | -1.65 | 0.10 |
| Odocoileus hemionus | 06:44 | 0.15 | 15689 | 20:56 | 0.24 | 2471 | -6.48 | <0.01 |

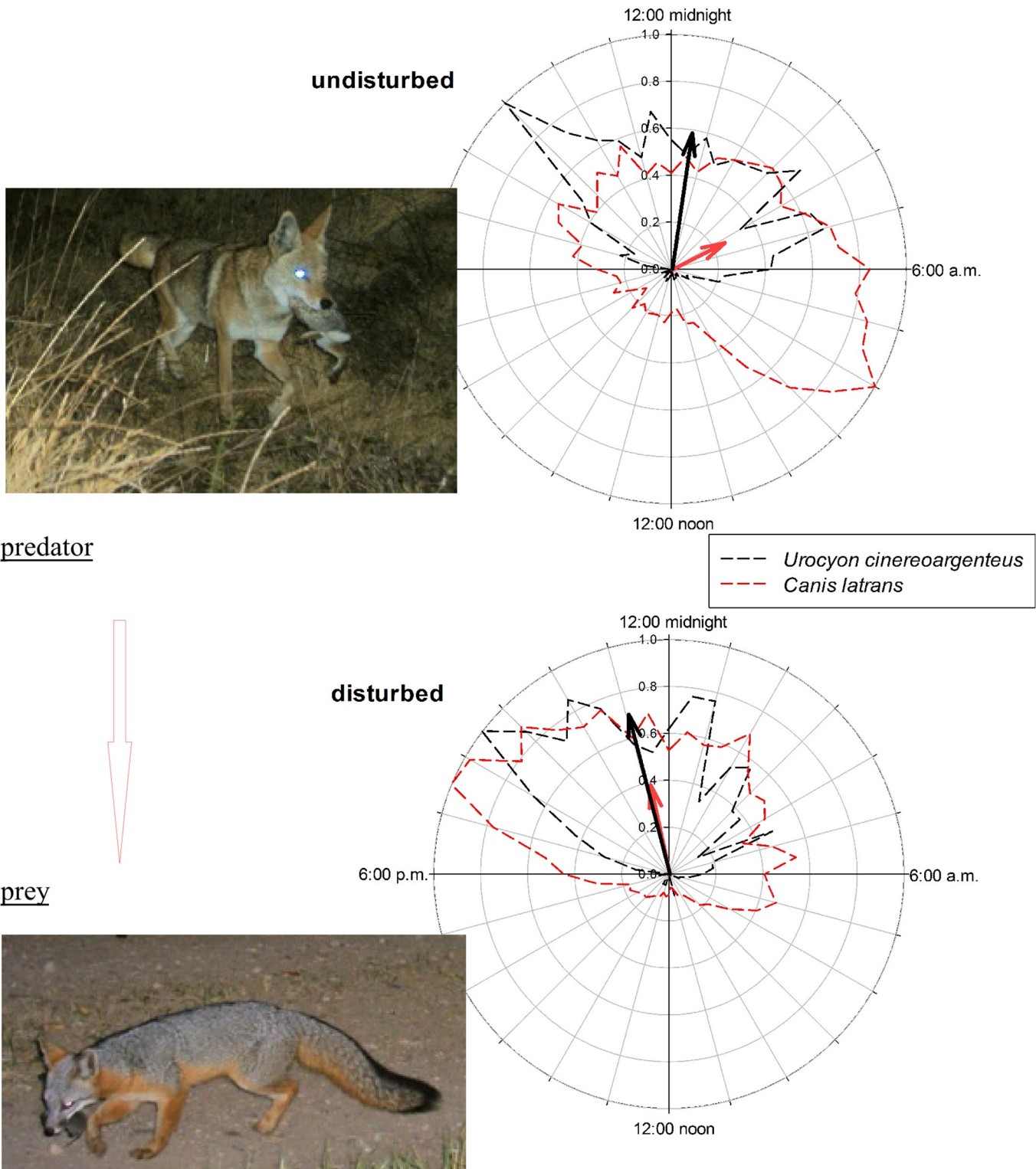

**Fig 1. Diel activity of the Gray Fox (*Urocyon cinereoargenteus*) and Coyote (*Canis latrans*) with or without human disturbance.** Arrows indicate time (direction) and proportional magnitude (length) of mean activity, and the "net" displays the spread of activity on a 24-h clock, binned at 30-min. intervals. Note the predator's (the Coyote) nocturnal shift when disturbance was present. The extent of overlap in activity increased from 33% to 45%.

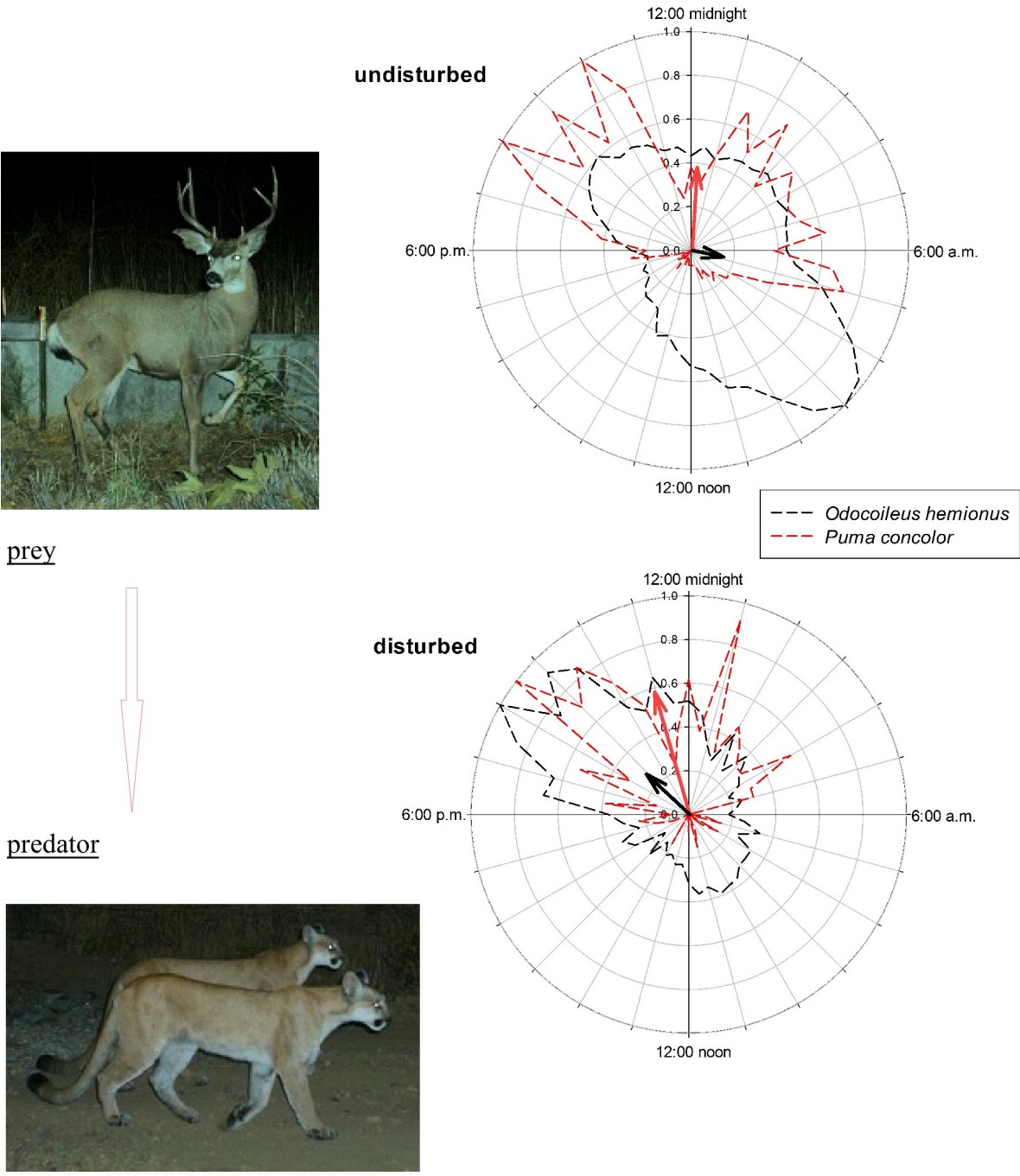

**Fig 2. Diel activity of the Mule Deer (*Odocoileus hemionus*) and Puma (*Puma concolor*) with and without human disturbance; see Fig 1 for an explanation of the graphs.** Note the prey's (the deer) nocturnal shift when disturbance was present. The extent of overlap in activity increased from 35% to 54%.

With probability 0.94, the true and underlying slope ($\beta_1$ = -0.13) of a beta regression of deer occupancy against year was negative, whereas with probability 0.87, the true and underlying slope ($\beta_1$ = 0.07) of a beta regression of Puma occupancy against year was positive (Fig 3). Also, the observed number of days on which both the Mule Deer and Puma were recorded at the

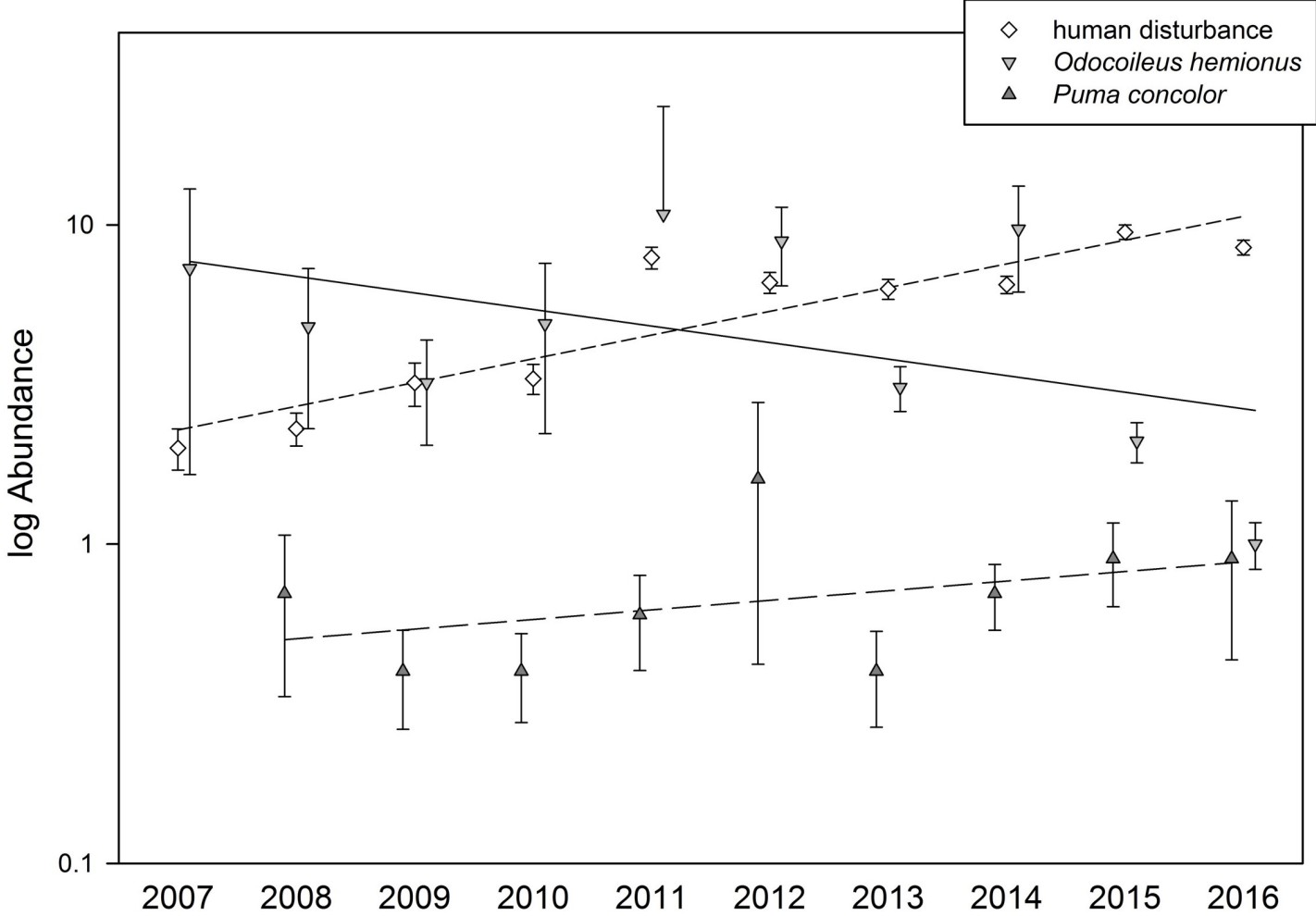

**Fig 3. Mean estimated abundance (from an *N*-mixture model, ±SE, and on a common log scale) of the Mule Deer (*Odocoileus hemionus*) and Puma (*Puma concolor*), as well as human disturbance for reference, with heuristic plots of mean slopes from a linear regression.** It is probable (Bayesian posterior $P = 0.94$) that deer occupancy ($\Psi$) has decreased over time whereas Puma occupancy has increased (Bayesian posterior $P = 0.87$). (Occupancy estimates for the Puma were unstable for 2007 and so abundance could not be estimated. Also, probability of detection was not correlated with year, although it may have increased slightly for the deer.).

same camera consistently was lower than an expectation if presence was independent of each other (Table 2).

## Discussion

Response of large and medium-sized mammals to human activity is difficult to characterize because it varies in time and space and with species and type of activity [36], yet an increasing body of literature suggests that the response is negative: in general, wild animals avoid humans, a pattern well established in coastal southern California [9,15–17]. A pattern is emerging, yet mechanisms by which animals avoid humans or come into conflict with humans often (but not always [37]) are not. Our data suggest that one mechanism involves diel shifts away from daylight hours, when human activity dominates—e.g., in our study area, nearly 95% (54,170 of 57,359 occurrences) of human disturbance was detected between 07:00 a.m. and 07:00 p.m. As a result, when disturbance was present the temporal distribution of two species, the Coyote and Mule Deer, shifted away from early morning to nearer the middle of the night (Figs 2 and

**Table 2. Presence of the MULE DEER (*Odocoileus hemionus*) and the PUMA (*Puma concolor*) across camera traps, with expected versus observed incidence of joint presence.** Data were aggregated by calendar day for the focal species alone (i.e., camera-trap records of species other than the deer and PUMA were not considered) and restricted to cameras that had at least one presence of the PUMA. Expected values were estimated from a joint probability under the assumption that the deer and PUMA occur independently, meaning observed joint (i.e., same day) presence much lower than expected implies that occurrence is not independent.

| camera | n | proportion of days with a presence | | joint probability | |
| | | Mule Deer | Puma | expected | observed |
| --- | --- | --- | --- | --- | --- |
| AG CH | 77 | 0.675 | 0.325 | 17 | 0 |
| AU TR | 509 | 0.986 | 0.022 | 11 | 4 |
| BL ST | 66 | 0.924 | 0.091 | 6 | 1 |
| BO SP | 618 | 0.989 | 0.013 | 8 | 1 |
| BO TR | 582 | 0.930 | 0.124 | 67 | 31 |
| CO MI | 182 | 0.967 | 0.038 | 7 | 1 |
| CO TR | 173 | 0.908 | 0.104 | 16 | 2 |
| DO CA | 106 | 0.830 | 0.179 | 16 | 1 |
| DO CA2 | 50 | 0.900 | 0.120 | 5 | 1 |
| DO CA3 | 29 | 0.897 | 0.103 | 3 | 0 |
| DR SP | 207 | 0.807 | 0.232 | 39 | 8 |
| DR SP2 | 149 | 0.926 | 0.121 | 17 | 7 |
| EA MW | 445 | 0.971 | 0.043 | 18 | 6 |
| EA MW2 | 41 | 0.976 | 0.024 | 1 | 0 |
| FR RO | 231 | 0.939 | 0.087 | 19 | 6 |
| FU BR | 204 | 0.941 | 0.069 | 13 | 2 |
| GY FO | 243 | 0.930 | 0.119 | 27 | 12 |
| LA RO | 319 | 0.937 | 0.085 | 25 | 7 |
| LI ME | 131 | 0.832 | 0.198 | 22 | 4 |
| LI ME2 | 35 | 0.771 | 0.286 | 8 | 2 |
| LI SI | 141 | 0.816 | 0.270 | 31 | 12 |
| LO WE1 | 121 | 0.992 | 0.008 | 1 | 0 |
| LO WE2 | 274 | 0.737 | 0.281 | 57 | 5 |
| ME SP | 10 | 0.800 | 0.200 | 2 | 0 |
| MO FR1 | 215 | 0.837 | 0.181 | 33 | 4 |
| MO FR3 | 178 | 0.781 | 0.247 | 34 | 5 |
| OV TR | 195 | 0.964 | 0.036 | 7 | 0 |
| RA TR | 77 | 0.987 | 0.026 | 2 | 1 |
| RO CA | 134 | 0.993 | 0.007 | 1 | 0 |
| RO CA2 | 35 | 0.857 | 0.171 | 5 | 1 |
| SO GY | 204 | 0.868 | 0.157 | 28 | 5 |
| UP WE | 133 | 0.932 | 0.075 | 9 | 1 |
| WE SP | 458 | 0.985 | 0.026 | 12 | 5 |
| WE TR1 | 791 | 0.991 | 0.059 | 47 | 40 |
| WE TR2 | 627 | 0.994 | 0.038 | 24 | 20 |
| WE TR3 | 359 | 0.997 | 0.028 | 10 | 9 |
| WE WI | 71 | 0.930 | 0.070 | 5 | 0 |

3). Such diel shifts may be commonplace [22], yet potential ecological implications of avoidance or conflict is little explored.

Our finding that relative risk of predation increases after human-induced diel shifts suggest important ramifications for predator–prey dynamics. We couch potential effects in terms of relative risk to draw a parallel between predation and disease transmission. In the latter, exposure to a disease is no guarantee of transmission, but as risk of exposure increases the

probability of contracting that disease increases. Likewise, increased risk of predation via diel shifts does not mean that a given individual will be killed, but that individuals' risk of mortality has increased. In this light, it is crucial to note that shifts we detected were not solely of species in one trophic level (e.g., secondary consumers); instead, we uncovered evidence for a shift in a primary consumer that brought it into greater temporal alignment with its chief predator (the MULE DEER into the PUMA's activity window) and for a shift in a secondary consumer that brought it into greater temporal alignment with prey (the COYOTE into the GRAY FOX's activity window). In either case the post-shift increase in relative risk can be seen as an increase in encounter rate. All else being equal, on the basis of Holling's disc equation—a means to estimate a predator's potential functional response to an increase in prey—increased encounter yields a higher rate of predation [38], which ultimately may depress prey populations.

Recent studies have underscored how altered predation regimes affect ecosystems [39–42]. We lack direct evidence for increased rates of predation in our system, yet indirectly we see MULE DEER decreases and PUMA increases coincident with increased human activity (Fig 3). And even if predation rates change little, increased presence of predators—or, indeed, increased *perceived* predation risk [39–40]—may nonetheless adversely affect prey species' behavior (e.g., habitat occupancy). For example, the risk of predation may itself be detrimental in that the mere presence of predators can decrease survival or fecundity of prey species [43], and temporal shifts of competitively or behaviorally dominant species, such as the COYOTE, may shrink niche space of subordinate species, such as the GRAY FOX [44]. Lastly, given that the COYOTE and GRAY FOX are potential prey for the PUMA, the COYOTE's shift to a more nocturnal pattern may further alter predator–prey dynamics relative to undisturbed systems.

We posit that subtle stress on prey populations that results from increased temporal overlap of predator and prey in combination with other population stressors, such as habitat loss, reduced habitat connectivity, and increased human activity [15–17], has ramifications for wildlife in urban wildlands where spatial and temporal refugia are limited. Human disturbance adds a stressor to fragmented landscapes that often support compromised habitats, and its effect may be unsustainable if not managed properly. It is possible that some prey find refuge from predators when humans are numerous because predators also avoid humans [26], but reserves in our study area are not large, so opportunity for spatial shifts is limited. In our system, human presence was as strongly negatively associated with the spatial and temporal activities of a primary consumer as it was for any of predators [17]. This effect was stark regardless of how we measured "disturbance": the pattern held whether humans were on foot, on bicycles, in vehicles, or on horseback or whether dogs accompanied them, the last a finding consistent with other work [45]. Further, we found that activity of the MULE DEER and COYOTE shifted temporally in a manner that increased probability of a predator–prey encounter in the absence of other avoidance behavior, adding a previously unrecognized stressor to wildlife prey species.

Despite the strong signal, our quantification may underestimate effects because we examined human presence rather than human abundance. Studies elsewhere have found that behavioral response of large mammals may not be apparent at low densities of humans and their accouterments but are as density increased [21,28]. Reserves in Orange County and other urban areas are visited by considerably more people than are protected areas in the studies cited above, so it may be that effects on wild animals of increased human density are stronger in urban areas. Yet access to reserves is important if we hope to foster a sense of stewardship and passion for natural habitats and their denizens, even if we must manage access carefully (and perhaps even strictly) if we hope to conserve natural habitats and their denizens. The majority of the protected landscapes in this study are under managed access and therefore generally subject to a clumped distribution of human activity and a lower level of overall activity

relative to neighboring parks that have 7-day access. As human use of parks increases, more research will be needed on both community-level and potential cascading effects of recreation [46] Further studies that test specifically for spatial effects of recreation on predator-prey dynamics, such as using a grid of cameras, could test the degree and scale of spatial impact on predator-prey behavioral shifts we suggest herein.

## Supporting information

**S1 Table. Specific locations of camera traps in coastal Orange County, California, 2007– 2016.**
(PDF)

**S1 Fig. Active cameras 2007–2015.**
(PDF)

**S2 Fig. Parameter estimates of peak activity of the Mule Deer (*Odocoileus hemionus*) and Coyote (*Canis latrans*) when camera trap images are excluded within 1 hour of the previous image of that same species or only within 5 minutes of the previous image.**
(PDF)

**S3 Fig. Temporal distributions for human activity in this study.**
(PDF)

**S4 Fig. Temporal distributions for "non focal" species in this study.**
(PDF)

## Acknowledgments

Special thanks to Irvine Ranch Conservancy staff members M. Fowler, D. Swenson, E. Sheehan, K. Brtalik, G. Salgado, A. Nelson, S. Lagarile, C. McCammon, R. Pratt, and S. Anon for their assistance with data gathering, and to UCSB Bren Intern Nico Alegria for help with data management. J. Paludi, A. and M. Fowler installed most cameras, and Y. C. Fang drafted maps. IRC volunteers, especially D. Newell, D. Millar, P. Wetzel, C. Carter, J. Hanson, D. Mahalik, and E. Hill, serviced wildlife cameras after installation. Our thanks to project team members W. Miller (USFWS), C. Beck (CDFW), J. Gump (OC Parks), and E. Boydston (USGS) for discussion and input and to members of the Patten lab (A. E. Adams, M. Dantzler-Kyer, D. C. Hille, E. A. Hjalmarson, G. Shahrokhi) for feedback on some ideas presented herein. We particularly appreciate a thorough and thoughtful review provided by W. Vickers of a pre-submission draft of the manuscript.

## Author Contributions

**Conceptualization:** Michael A. Patten.

**Data curation:** Jutta C. Burger.

**Formal analysis:** Michael A. Patten.

**Funding acquisition:** Milan Mitrovich.

**Methodology:** Jutta C. Burger.

**Project administration:** Milan Mitrovich.

**Resources:** Jutta C. Burger.

**Writing – original draft:** Michael A. Patten.

**Writing – review & editing:** Jutta C. Burger.

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
