## [Decision Letter · Decision Letter 0]

10 Jul 2019

PONE-D-19-14511

The Intersection of Human Disturbance and Diel Activity, with Potential Consequences on Trophic Interactions

PLOS ONE

Dear Dr. Patten,

Thank you for submitting your manuscript to PLOS ONE. After careful consideration, we feel that it has merit but does not fully meet PLOS ONE’s publication criteria as it currently stands. Therefore, we invite you to submit a revised version of the manuscript that addresses the points raised during the review process.

Both reviewers found that this study is valuable and well written, so they provided constructive comments to improve the MS. However, both referees manifested that methodology and statistical analyses need to be reviewed carefully. Then, please check your MS throughly to asses referees' comments, doubts, and suggestions, paying attention to following issues: human activity patterns (disturbed vs undisturbed must be clarified); diel patterns you found should be related at temporal or spatial level; sharing space between prey and predators; the use of peak activity as measurement; analyses using abundance time series.

We would appreciate receiving your revised manuscript by Aug 24 2019 11:59PM. To enhance the reproducibility of your results, we recommend that if applicable you deposit your laboratory protocols in protocols.io, where a protocol can be assigned its own identifier (DOI) such that it can be cited independently in the future. For instructions see: http://journals.plos.org/plosone/s/submission-guidelines#loc-laboratory-protocols

We look forward to receiving your revised manuscript.

Kind regards,

Paulo Corti, Ph.D.

Academic Editor

PLOS ONE

**Journal Requirements:**

2.  In your Methods section, please provide additional location information of study area, including geographic coordinates for the data set if available.

**Comments to the Author**

1. Is the manuscript technically sound, and do the data support the conclusions?

Reviewer #1: Yes

Reviewer #2: Partly

2. Has the statistical analysis been performed appropriately and rigorously? 

Reviewer #1: Yes

Reviewer #2: No

3. Have the authors made all data underlying the findings in their manuscript fully available?

Reviewer #1: No

Reviewer #2: Yes

4. Is the manuscript presented in an intelligible fashion and written in standard English?

Reviewer #1: Yes

Reviewer #2: Yes

5. Review Comments to the Author

Reviewer #1: Main comments to authors

The authors analyze a very large data set of large-mammal pictures taken by authomatic cameras in protected areas in California. The authors look at temporal patterns of activity to analyze if human presence is coupled with a shift in anuimal activity, and if this kind of changes can change predator-prey and agonisitic behaviour/intra-guild predation risks. Data are robust and methods in general appropriate, though some improvements are needed. See more detailed comments below.

- Please, include in results the diel activity graph of human activity. It is central to your discussion and readers only know that human activity is 95% diurnal. Also, indicate the percentage of disturbed vs. Undisturbed days of your cameras, and the variability among them.

- The methodological observation in l. 120-125. pinpoints something rather relevant that has to be taken into account, discussed and potentially analyzed in the paper. The consistency of results either defining the ‘disturbed/undisturbed’ as presence of human activity before or after the animal detection shows that it is not the case that the (recent) presence of human disturbance shapes the response you describe in the paper. Thus, the differences in diel patterns you find must derive from a correlation/response at a higher level, either temporal or spatial.

In the case that there is ‘temporal correlation in your observations’ (the same camera takes repeatedly pictures from the same animal species with different beahaviours -disturbed/undisturbed-, but observation of the former and latter are aggregated), your results show that the behaviour of animals changes seasonally in sites visited by humans during one season (e.g. Easter vacation, summer…). That’s why you find that it is the same to analyse data looking to the previous or the next day (most probably both are high- or low-human occupancy)

The other option (spatial correlation) is that you always get animals in ‘disturbed’ conditions in some points (e.g. cameras located closest to parking lots at entrances) and ‘undisturbed’ animals always in other sites (deeper in the reserve where fewer hikers reach). In this case, you may have individuals that always behave in one of the two conditions.

The above does not invalidate your research at all, but it is rather important to know about it to understand the potential implications. I suggest analysing the spatial and temporal distribution of disturbance as well as those of animal observations, and including descriptive data on that in the paper to properly discuss the issue. In a (somewhat) similar vein, I suggest informing also the frequency of cameras with detections of species in pairs for a somewhat narrower time window (e.g. number of coincidences in season and site of puma-Odocoileus). All this information can be very valuable and worth discussing.

- l. 125-129. I suggest revising the use you do (in results and discussion) to the concept of ‘peak activity’ and ‘change in peak activity’ due to the statistics involved. The Rayleigh’s test formally gives you the presence of directionality in the clock vs. randomness, but it is not fully informative of ‘a peak hour’ (see Odocoileus case in Figure 2a for a clear case of ‘peak from a bi-modal distribution’). The point with this test is that it computes an average vector for all observations and analyses if its length is larger than expected under random conditions (some bi-modalities may even show lack of peak!!). The second analysis based on ranks (it is not fully explained but from my experience with circular statstics I guess the details) also looks for differences in ‘temporal distribution’ rather than differences in ‘peak’. As a conclusion, I suggest better dealing with it as changes in temporal distributions.

- l. 148-150. Do not include the option for ‘spatial avoidance’ since you can’t say about that with this approach: In case prey spatially avoided predators, then their presence in other cameras would compensate for this movement. The case that they ‘move away’ of your cameras (but they stay in the area) can’t be discussed on scientific grounds (you would need a posteriori explanations based on imperfect sampling with your cameras added to a biased presence of predators close to your cameras). Moreover, with your data you can’t blame changes in diel activity as responsible in population abundances. You just describe a reduction in deer numbers occurring at the same time that puma numbers increase, but we do not have any clue about the reason (ok, it can be increased predation, but higher numbers of puma simply predate more to survive and we are not informed if diel activity of deer has gradually changed since 2007 until 2016, or about changes in attack success of puma along the years)

- Methods in l. 143-153need to be clarified (and/or re-organized). I understand (right? wrong? And readers should not face many doubts) that you first did a kind of Bayesian occupancy modelling to estimate probability of detection and presence using raw 0-1 data. From that you extract the slopes –interannual change- (those painted in fig. 3? Why don’t you include a second y-axis scaled accordingly?) and their plausabilities of being different from 0. After you have the detection probabilities for each year, you use the raw data of number of detections per quarter per camera corrected for detectability to compute abundance data. Finally, it is not clear if lines in fig 3 are just regression lines fitted to the (log-)abundance data and drawn ignoring any kind of their fit statistics. In short: explain it a bit better and in the logical order of construction (I may have passed several issues).

- I encourange including (it can be as supplementary) the diel activity graphs for all species in both situations (one figure by species).

Other comments

- Include a map of the study area with the distribution of protected areas-working sites and urban areas (not needed in a very precise detail that may compromise the robbery of cameras)

- l. 108-111. Further justify why you do not use the 1-h lapse time between animal snaps to differentiate ‘animal visits’ since it is the time window usually applied in studies of carnivores. Moreover, two observations of the same species in the same place in 5-10’ when you work with large mammals (puma, deer) will correspond in most cases to the same individual just moving around (e.g. puma) or feeding (deer). On the contrary, you applied the one-hour rule for people. It would be also needed to know how many observations you have that do not fit the one-hour rule but only the 5’ one.

- Figures 1 and 2 apparently show data by 30’ intervals. If this is the case (in order to somewhat smooth their appearance) explain it. In the caption, substitute ‘…relative to human disturbance’ by something like ‘… under the disturbance of human and its absence’. The present captions give the idea of confronting in the same plate diel activity of the animals and human activity.

- You can easily compute the percentage of overlap shown in figures 1 and 2, and include in in results. This features will have the advantage of being rather easy to understand for readers.

Typographical errors and other minor issues

- l. 137-139. In fact you compute ‘the odds of two odd ratios’. I do not know if it could be better to find another name or explain it better (it is a suggestion, it first surprised me and I took a bit time to understand it… but I am not English native)

- l. 192 … potential increase in encounter rate

- Table 1. Do not include second in the peak time. Use the same number of decinal positions in all ‘P’ values

Reviewer #2: Patten et al. use a long-term camera trapping dataset to evaluate shifts in species diel activities in response to human disturbance and assess the consequences for predator-prey relationships. The question being addressed – the influence of humans on ecological interactions – is important, and the data used are extensive. The manuscript is clearly written overall.

However, I have three major concerns with this manuscript:

1) A prey species can share the same diel pattern that its predator, yet avoid the latter in space. This mechanism is hinted at in the Puma-Mule Deer analysis, but I am not convinced by that analysis (see point 2 below). Until the authors show that these diel shifts are or aren’t accompanied by changes in space use, conclusions regarding changes in predation risk are unfortunately not supported. Unless I have missed something, I do not see this kind of analysis in the manuscript.

2) Using the “peak activity” as the sole metric with which to compare activity distributions seems a bit restrictive. What if the distribution is bimodal, hypothetically?

3) The analysis of abundance time series could be improved. First, given the cyclic and non-independent nature of the time series, I’m not sure a simple linear regression is the most appropriate approach to quantify trends. On top of this, there is added variation due to seasonality/quarter that does not seem to be accounted for. Second, why not implement a single model assessing prey abundance as a function of predator abundance, human disturbance, with random effects to account for year and quarter. The temporal unit would be the quarter. Another approach could be to look at cross-correlations (possibly lagged) between the time series considered. Overall, I admit that this analysis was a bit unclear to me.

Minor comments:

l. 28 – not clear what “avoidance can mirror predation” means here. Could you please clarify or reword?

l. 52 – “basic ecological terms” is a bit vague. What would these be?

l. 92 – Cuddeback instead of “Cuddleback”

l. 108-11 – I think the difference in independence thresholds needs justifying.

l. 117-9 - why wasn’t the disturbance caused by the researchers setting/checking the cameras also included?

l. 143 – a reference to support this statement is needed

l. 164-66 – use of the present tense is awkward, I suggest changing to past.

l. 166 – perhaps use “puma” instead of “lion” to maintain consistency.

l. 172-174 – Awkward sentence, I suggest rewording.

l. 192 – not if space use shifts as well (see major concern 1)

6. PLOS authors have the option to publish the peer review history of their article (what does this mean?). If published, this will include your full peer review and any attached files.

Reviewer #1: No

Reviewer #2: Yes: Jeremy Cusack

---

## [Author Response · Author response to Decision Letter 0]

15 Oct 2019

Please see the cover letter, in which, and to the best of ability, I addressed all reviewer comments and critiques.

---

## [Editor Report · Decision Letter 1]

29 Oct 2019

PONE-D-19-14511R1

The Intersection of Human Disturbance and Diel Activity, with Potential Consequences on Trophic Interactions

PLOS ONE

Dear Dr. Patten,

Thank you for submitting your manuscript to PLOS ONE. After careful consideration, we feel that it has merit but does not fully meet PLOS ONE’s publication criteria as it currently stands. Therefore, we invite you to submit a revised version of the manuscript that addresses the points raised during the review process.

I have revised your answers to both reviewers and most of them fulfilled their requests. However, I believe some of them still need some work, so here what I think is necessary to take into account to improve your manuscript:

- Line 108-111. Please exclude the 15% from your data set or show the analysis where is probed that results are not affected when the 15% that doesn't fulfill the 1 h rule is included in the analyses.

- Animal common names should be in small caps.

- Please provide citations to back up information in between lines 149 and 154, and also in 165-167.

- In lines 259-262 Please, avoid redundancy and speculation, so rewrite these sentences.

I think it is importan to correctly answer this request from reviewer #1:

Requests: You can easily compute the percentage of overlap shown in figures 1 and 2, and include in in results. This features will have the advantage of being rather easy to understand for readers.

Response: I do not feel it is easy to calculate overlap, apart from program ‘overlap’ in R, which is designed to do just that. I have concerns about this package, though, because it does not account for the circular nature of the data, and hence I have not reported results from it. I still do not feel this paper is the place to air concerns about that approach, particularly because Fig. 1 and Fig. 2 “speak for themselves,” as the old trope goes.

Please reduce speculations so show the percentage of change and also there were differences between treatments.

We would appreciate receiving your revised manuscript by Dec 13 2019 11:59PM. To enhance the reproducibility of your results, we recommend that if applicable you deposit your laboratory protocols in protocols.io, where a protocol can be assigned its own identifier (DOI) such that it can be cited independently in the future. For instructions see: http://journals.plos.org/plosone/s/submission-guidelines#loc-laboratory-protocols

We look forward to receiving your revised manuscript.

Kind regards,

Paulo Corti, Ph.D.

Academic Editor

PLOS ONE

---

## [Author Response · Author response to Decision Letter 1]

25 Nov 2019

- Line 108-111. Please exclude the 15% from your data set or show the analysis where is probed that results are not affected when the 15% that doesn't fulfill the 1 h rule is included in the analyses.

RESPONSE: I did the latter; see S2 Fig.

- Animal common names should be in small caps.

RESPONSE: Okay.

- Please provide citations to back up information in between lines 149 and 154, and also in 165-167.

RESPONSE: We fear a misunderstanding here. Lines 149–154 only restate basic mathematical definitions of joint probability and odds in terms of our data. On lines 165–167 we proffered a specific prediction as opposed to a relationship from the literature. To avoid further confusion, we changed “expected” to “predicted.”

- In lines 259-262 Please, avoid redundancy and speculation, so rewrite these sentences.

RESPONSE: We rewrote the closing of the final paragraph.

- I think it is importan to correctly answer this request from reviewer #1: Requests: You can easily compute the percentage of overlap shown in figures 1 and 2, and include in in results. This features will have the advantage of being rather easy to understand for readers. Response: I do not feel it is easy to calculate overlap, apart from program ‘overlap’ in R, which is designed to do just that. I have concerns about this package, though, because it does not account for the circular nature of the data, and hence I have not reported results from it. I still do not feel this paper is the place to air concerns about that approach, particularly because Fig. 1 and Fig. 2 “speak for themselves,” as the old trope goes.

Please reduce speculations so show the percentage of change and also there were differences between treatments.

RESPONSE: Reviewer comments to the contrary, computing overlap is not easy, and I do not wish to air publicly my concerns about the ‘overlap’ package in R. I nonetheless understand the desire to see how % overlap changed. I accomplished this by porting graphs into ImageJ, a shareware package, to estimate overlap area (A *intersection* B) and total area (A *union* B). Their quotient approximates proportion of overlap, which I present in the text. As for “differences between treatments,” that is addressed already in the odds ratio analysis, but to supplement that analysis I wrote code to obtain a Bayesian analog to the Watson–Wheeler two-sample test of difference in mean angle. I report those results now, too.

---

## [Editor Report · Decision Letter 2]

27 Nov 2019

The Intersection of Human Disturbance and Diel Activity, with Potential Consequences on Trophic Interactions

PONE-D-19-14511R2

Dear Dr. Patten,

We are pleased to inform you that your manuscript has been judged scientifically suitable for publication and will be formally accepted for publication once it complies with all outstanding technical requirements.

With kind regards,

Paulo Corti, Ph.D.

Academic Editor

PLOS ONE

---

## [Editor Report · Acceptance letter]

3 Dec 2019

PONE-D-19-14511R2 

The Intersection of Human Disturbance and Diel Activity, with Potential Consequences on Trophic Interactions 

Dear Dr. Patten:

I am pleased to inform you that your manuscript has been deemed suitable for publication in PLOS ONE. Congratulations! Your manuscript is now with our production department. 

With kind regards,

on behalf of

Dr. Paulo Corti 

Academic Editor

PLOS ONE